# Insomnia, Daytime Sleepiness, and Quality of Life among 20,139 College Students in 60 Countries around the World—A 2016–2021 Study

**DOI:** 10.3390/jcm12020692

**Published:** 2023-01-15

**Authors:** Mateusz Babicki, Patryk Piotrowski, Agnieszka Mastalerz-Migas

**Affiliations:** 1Department of Family Medicine, Wroclaw Medical University, 51-141 Wroclaw, Poland; 2Section of Epidemiology and Social Psychiatry, Department and Clinic of Psychiatry, Wroclaw Medical University, 50-367 Wroclaw, Poland; 3Division of Consultation Psychiatry and Neuroscience, Wroclaw Medical University, 50-367 Wroclaw, Poland

**Keywords:** insomnia, daytime sleepiness, human development index, gross domestic product per capita, quality of life

## Abstract

Background: Sleep disorders are a widespread phenomenon, and the number of individuals suffering from them is increasing every year, especially among young adults. Currently, the literature lacks studies that cover both countries with different levels of development and a period before the announcement of the ongoing COVID-19 pandemic. Therefore, this study aims to globally assess the prevalence of insomnia and daytime sleepiness among students and assess their quality of life. Methods: For this purpose, our own questionnaire was distributed online via Facebook.com. In addition to the questions that assessed socioeconomic status, the survey included psychometric tools, such as the Athens insomnia scale (AIS), the Epworth sleepiness scale (ESS), and the Manchester short assessment of the quality of life (MANSA). The survey distribution period covered 31 January 2016 to 30 April 2021. Results: The survey involved 20,139 students from 60 countries around the world. The vast majority of the students were women (78.2%) and also those residing in countries with very high levels of development and/or high GDP (gross domestic product) per capita at 90.4% and 87.9%, respectively. More than half (50.6%) of the respondents (10,187) took the survey before the COVID-19 pandemic was announced. In the group analyzed, 11,597 (57.6%) students obtained a score indicative of insomnia and 5442 (27.0%) a score indicative of daytime sleepiness. Women, low-income residents, and nonmedical students were significantly more likely to have scores indicating the presence of insomnia. Individuals experiencing both sleepiness (B = −3.142; *p* < 0.001) and daytime sleepiness (B = −1.331; *p* < 0.001) rated their quality of life significantly lower. Conclusions: Insomnia and excessive daytime sleepiness are common conditions among students worldwide and are closely related. The COVID-19 pandemic significantly altered students’ diurnal rhythms, which contributed to an increase in insomnia. Students in countries with a high GDP per capita index are significantly less likely to develop insomnia compared to the residents of countries with a low GDP per capita index. Sleep disorders definitely reduce the quality of life of students.

## 1. Introduction

The spectrum of sleep disorders is very broad and can include but is not limited to dyssomnias, insomnia, pathological sleepiness, narcolepsy, parasomnias, or sleep (hypnagogic and hypnopompic) hallucinations. Some of the most common conditions in the general population include, among other things, insomnia and daytime sleepiness [1]. Insomnia is a medical condition in which a person has difficulty falling asleep or staying asleep [2]. On the other hand, daytime sleepiness is defined as difficulty in maintaining appropriate levels of wakefulness [3]. The two conditions have a close relationship, and it was proved that those suffering from insomnia show higher levels of daytime sleepiness [3,4]. According to global reports, every year, there are more and more individuals suffering from sleep disorders, especially young people [5,6]. A specific group is young adult students who, due to a series of life changes that are associated with studying (change of residence, greater independence, as well as increased consumption of stimulants, such as alcohol, cannabinoids, or psychostimulants), are far more likely to develop sleep disorders [7,8]. Young people rarely follow basic sleep hygiene rules, which have a huge effect on sleep quality [9]. Low knowledge of sleep hygiene among college students was also demonstrated. Furthermore, they adhered to the basic principles of sleep hygiene to a low degree [10,11]. Moreover, it is believed that young adults may still function properly within the late sleep phase and, for this reason, they stay awake longer, go to bed late, and have to get out of bed early due to their daily responsibilities. This leads to a situation where the amount of sleep becomes insufficient [12]. These changes were particularly observed among college students [13]. Furthermore, reference should also be made to the ongoing COVID-19 pandemic and its direct impact on mental health, including sleep. Numerous studies indicate that this situation has resulted in an increase in insomnia, especially among young people [14,15]. It was also observed that the ongoing COVID-19 pandemic significantly altered the previous sleep patterns among college students, who went to bed late, got out of bed late, and had numerous naps during the day. All this leads to a deterioration of sleep quality, which is the most important guarantor of its efficiency. The reasons are found in the introduction of lockdowns, remote working and learning, and significant restrictions on social interaction [16].

The role of sleep is invaluable, and it is believed that it is one of the fundamental needs of humans. Sleep ensures the maintenance of both mental and physical health [17]. It is known that those suffering from sleep disorders are far more likely to suffer from chronic diseases, such as hypertension, diabetes, obesity, and death [18]. Sleep disorders were found to increase the risk of developing mental disorders, including depression, anxiety, and personality disorders. On the other hand, sleep disorders can also be a symptom of a developing mental health condition or its exacerbation [19]. Moreover, poor sleep has a direct impact on daily life. It is associated with lower performance at school and work and with concentration and memory difficulties [2,20,21]. A correlation between psychoactive substance use and sleep was also proven. The dependency mechanism is complex and works in two directions. Users of alcohol, cannabinoids, or psychostimulants have problems with their sleep physiology; on the other hand, those with poor sleep quality are far more likely to turn to stimulants in the hope of improving it [22,23,24].

Adequate sleep was also found to have a huge effect on the assessment of quality of life (AQoL) index, which has become increasingly important in recent years. AQoL, in addition to a biological health assessment, is an essential part of a comprehensive patient assessment [25]. The AQoL is made up of many factors, such as mental health, physical health, social relationships, economic status, etc., and thus its measurement is not straightforward. To this end, several standardized psychometric tools have been developed. Their questions address various aspects of life, thus enabling it to be reliably assessed [26]. Several studies revealed that both organic and mental health conditions contribute significantly to a reduced QoL, which can also be used for identifying a range of health problems. These problems can have an impact on patients, and their modification will improve patients’ life [27,28].

There can be considerable difficulty in terms of the diagnostic evaluation of sleep disorders such as insomnia and sleepiness, which are very complex. The most accurate tools are instrumental methods, especially polysomnography. Due to the methodologies of this study, however, it was not feasible to use polysomnography as a screening tool. Therefore, appropriate questionnaires with a high level of sensitivity and reproducibility, when compared to instrumental methods, were developed, and they can also be used in population studies [29,30].

Currently, the literature is rich in studies on sleep disorders; however, they focus on a specific time and place. There is no single study that covers the different regions of the world with different cultural backgrounds, economic statuses, and levels of development. Socioeconomic status has long been proven to have a direct effect on health by affecting living conditions and behavioral patterns, especially health ones. Despite this, their effects on sleep are still poorly understood, and the data are not consistent and need to be standardized [31,32].

Therefore, this study aims to globally assess the prevalence of insomnia and daytime sleepiness among college students and assess their QoL. This study explored the relationships between sleep, QoL, and stimulants, such as alcohol, cannabinoids, psychostimulants, or anxiolytics. Furthermore, it explored the differences in terms of AQoL, the prevalence of insomnia, and daytime sleepiness according to socioeconomic status, as measured by GDP (gross domestic product) per capita and the human development index (HDI).

## 2. Materials and Methods

### 2.1. Participants

Table 1 shows the characteristics of the study group. The survey was completed by 20,431 respondents. A total of 292 people did not agree to participate in the survey and/or were not students. Finally, 20,139 respondents from 60 countries around the world with a mean age of 22.6 ± 3.6 years were included in the study. The vast majority of respondents were women (78.2%), nonmedical students (77.6%), and first-year university students (27.3%). Furthermore, students residing in countries with very high levels of development and/or high GDP per capita accounted for 90.4% and 87.9%, respectively. More than half (50.6%) of respondents (10,187) took the survey before the COVID-19 pandemic was announced. The most commonly used stimulant among students was alcohol. A total of 82% of students consumed alcohol at least once within the 3 months prior to the survey. Fewer than 25 percent (22.6%) of students admitted to using sedatives/hypnotic drugs, with an upward trend observed following the announcement of the COVID-19 pandemic.

### 2.2. Assessment

The CAWI (computer-assisted web interview) survey was designed based on our own questionnaire that was distributed online via a social networking site. A convenient sampling method was used in the selection of the study sample. The questionnaire was distributed to student groups around the world. The survey was fully anonymous, and voluntary, and its participants were free to opt-out at any stage of the survey without giving any reason. Prior to participating in the survey, respondents were informed of the aims and methodology of the study. Subsequently, they gave their informed consent to participate in the study. Additionally, respondents had to confirm their student status. In the case of both positive answers, the respondents proceeded to the relevant part of the questionnaire. If there was any negative answer, the survey was automatically terminated. The survey distribution period was from 31 January 2016 to 30 April 2021.

Inclusion criteria included being a college student and giving informed consent. In contrast, the exclusion criteria included: lack of college student status, lack of consent to participate in the study, and age of <18 years old.

The study was approved by the Bioethics Committee of the Wroclaw Medical University and was conducted in accordance with the Declaration of Helsinki.

The authors’ own questionnaire consisted of two parts. The first part involved sociodemographic data, including age, sex, country of residence, and study data: year of study and university profile (medical/non-medical). Moreover, the consumption of stimulants, such as alcohol, cannabinoids, psychostimulants, and sedatives/hypnotic drugs, was assessed within the last three months prior to participation in the survey. The second part of the survey consisted of standardized psychometric tools, such as AIS, ESS, and MANSA.

AIS (Athens insomnia scale) is an 8-item tool for assessing insomnia. AIS questions are based on the ICD-10 criteria for insomnia and a 4-point Likert scale (0—no difficulty sleeping; 3—severe difficulty). The tool assesses difficulty falling asleep, waking up at night, waking up in the morning, total sleep duration, sleep quality, mood the following day, mental and physical health the following day, and sleepiness during the day. The analysis of the tool is based on a summary score, and the cut-off point is 6. This tool has high sensitivity (93%) and specificity (85%). A high internal consistency for Cronbach’s alpha was found −0.827 [33,34];

ESS (Epworth sleepiness scale) is a tool for assessing daytime sleepiness based on 8 questions that assess the likelihood of falling asleep in specific situations. The maximum number of points is 24, and the cut-off point is 11. Furthermore, for values above 15, pathological sleepiness may be suspected. The reliability of the tool, as measured by Cronbach’s alpha coefficient, was 0.742 [35,36,37];

MANSA (Manchester short assessment of quality of life) is a 16-item tool for the subjective AQoL. This tool assesses satisfaction with, among other things, mental health, physical health, leisure activities, and relationships with family and friends. A maximum of 92 points can be scored using this tool; however, the higher the score, the better the AQoL. The internal consistency of the tool was 0.764 [38,39].

Then, based on the survey completion period, respondents were divided into the period before the announcement of the COVID-19 pandemic (until 11 March 2020) and the period during the pandemic—from 11 March 2020 onwards, according to the date of its announcement by the WHO [40]. Moreover, based on the country of residence, respondents were distributed by GDP per capita and HDI based on data from the World Bank and the United Nations Development Programme (UNDP). The assessment of HDI identified groups with very high, high, medium, and low development rates. In assessing the GDP per capita index, the following groups were distinguished: high, upper-middle, lower-middle, and low income [41,42].

### 2.3. Statistical Analysis

The analysis pertained to qualitative, quantitative, and dichotomous variables. Basic descriptive statistics methods were used for the quantitative variables. The chi-squared test was used to assess significant differences in terms of demographics and the psychoactive substances used among students regarding the COVID-19 pandemic. The age difference between the pandemic stages was assessed using the t-test. Kendall’s tau correlations were used for assessing the level of correlation between individual scale questions and the final scale score. A complex backward stepwise logistic regression model was constructed to determine the influence of factors on the risk of developing insomnia, where the dependent variable was the analysis of the AIS scores (insomnia/no insomnia) and the endogenous variables were sociodemographic data (age, sex), year of study, university profile, COVID-19 pandemic, HDI distribution, GDP per capita distribution, and psychoactive substance use (alcohol, cannabinoids, psychostimulants, hypnotic drugs). An analogous model was built to assess daytime sleepiness. In this case, the dependent variable was the ESS (sleepiness/no sleepiness) analysis. Subsequently, a complex backward stepwise linear regression model was constructed to assess the effects of sociodemographic variables, year of study, university profile, COVID-19 pandemic, HDI distribution, GDP per capita distribution, psychoactive substance use, insomnia, and daytime sleepiness on AQoL.

Statistica 13.3 software was used for the calculations.

In each case, *p* < 0.05 was considered statistically significant.

## 3. Results

### 3.1. Sleep Disorders and Risk Factors

In the group analyzed, the mean AIS score was 8.26 ± 4.35, within which 11,597 (57.6%) students had a score indicating the presence of insomnia. An analysis of the individual questions included in the scale revealed that 73.6% of students describe their sleep duration as insufficient, and 71.2% rate their sleep quality as unsatisfactory. More than one-third (78.5%) of students reported waking up during the night, and 78.9% had difficulty falling asleep. In terms of the questions analyzed, it was sleep quality that was most strongly associated with the final AIS score (r = −0.486, *p* < 0.001) rather than sleep duration (r = 0.417, *p* < 0.001). Regarding potential risk factors, women were found to have a higher risk of developing insomnia (OR 1.25; 95%Cl 1.67, 1.35; *p* < 0.001). Furthermore, it was shown that the risk of developing insomnia decreases with subsequent years of study. Medical students had a lower risk of developing insomnia (OR 0.90; 95%Cl 0.84, 0.97; *p* = 0.004) compared to nonmedical students. There were no statistically significant differences between the students according to the development levels of the country they were in. In contrast, a negative correlation was shown between GDP per capita and insomnia. The pandemic showed a more than two-fold increase in the risk of students developing insomnia.

For the Epworth scale (ESS), the mean score obtained by the students was 7.90 ± 4.23, and 5442 (27.0%) students obtained a score indicating the presence of daytime sleepiness. The most common situations in which respondents indicated the likelihood of falling asleep included afternoon rest (90.4%), riding the bus (73.4%), and watching TV (72.5%). Moreover, 10.4% of students indicated, with varying degrees of likelihood, falling asleep while driving a car. A risk factor analysis found that the risk of daytime sleepiness decreased after the pandemic outbreak. The use of hypnotic drugs was found to have a definite negative impact on both the risk of developing insomnia and daytime sleepiness. Psychostimulants significantly increase the risk of both insomnia (OR 1.51; 95%Cl 0.64, 1.89; *p* < 0.001) and excessive daytime sleepiness (OR 1.14; 95%Cl 1.04, 1.26; *p* < 0.001).

A detailed comparison of the risk factor analysis for the development of insomnia and daytime sleepiness is shown in Table 2.

### 3.2. The Assessment of Quality of Life (AQoL)

The mean score obtained in the analysis of MANSA was 60.9 ± 11.46. The linear regression analysis revealed that the subjective AQoL scores were higher during the COVID-19 pandemic than before it was announced (*p* < 0.001). The assessment of the HDI showed that the residents of countries with a very high HDI score rated their QoL at the highest level (B = 2.001; *p* = 0.002), while no differences were shown in terms of the GDP per capita distribution. There was a negative correlation between the use of hypnotic drugs, psychostimulants, and quality of life. The QoL of those individuals experiencing both sleepiness (B = −3.142; *p* < 0.001) and daytime sleepiness (B = −1.331; *p* < 0.001) was rated significantly lower.

A detailed comparison of the MANSA scores is shown in Table 3.

The analysis of individual MANSA questions revealed that students suffering from insomnia and daytime sleepiness in each question scored lower on average than their peers with normal sleep. In terms of the subjective assessment of life satisfaction, students suffering from insomnia scored, on average, 0.77 points lower (*p* < 0.001). The students also rated their mental and physical health significantly lower in the presence of insomnia and daytime sleepiness (*p* < 0.001). A detailed comparison is shown in Appendix A.

### 3.3. Correlation between Individual Scales

The correlation analysis between the individual scales showed that the higher the mean scores for both AIS and ESS, the lower the AQoL scores (rAIS = −0.355; *p* < 0.001; rESS = −0.155; *p* < 0.001). Moreover, the two scales have a close relationship, and the higher the AIS score, the higher the ESS score (r = 0.153; *p* < 0.001).

## 4. Discussion

Sleep disorders are a common phenomenon that occurs all over the world, and every year the number of individuals suffering from these disorders increases. Despite the huge number of studies regarding sleep, few studies are conducted on a long-term basis and cover countries with different cultural and socioeconomic backgrounds. Therefore, this study primarily aims to assess the prevalence of insomnia and daytime sleepiness according to countries with different levels of development and wealth. Also, the purpose of this study is to assess their impact on students’ QoL and search for potential risk factors for the development of sleep disorders, including the ongoing COVID-19 pandemic. To the best of the authors’ knowledge, this study is one of the first in the world with this kind of scope and study group size, covering a period both before and during the COVID-19 pandemic.

The results of this study clearly indicate that sleep disorders are a serious health problem among students. According to the analysis of the AIS, 11,597 (57.6%) respondents obtained a score indicating insomnia, and 5442 (27.0%) respondents scored 11 points or more on the ESS, which indicates excessive daytime sleepiness. Insomnia is identified in many research reports as the most common sleep dysfunction among students, and its prevalence varies according to the population surveyed, the period of data collection, and the tools used [43]. For example, in the period from 2015–2018, the average prevalence of insomnia among students in South Asia (India, Pakistan, Nepal, and Bangladesh) was 52.1%, with a range from 35.4% to 70% [44]. During the same period, in a study by Haile et al., the prevalence of insomnia among Ethiopian students was 61.6% [45]. In contrast, in a study among Sudanese students, the percentage of students with sleep problems was as high as 82.5% [46]. During the same period, the insomnia severity among students from affluent countries was 37.2, 30.5, 19.7, and 7.7% for Chinese, Norwegian, Polish, and German students, respectively [47,48,49,50]. These data imply that the place of residence, including socioeconomic status, can have a tremendous impact on sleep. This is also supported by studies using instrumental tools (actigraphy and polysomnography) and selfassessment questionnaires, where students with lower socioeconomic status were found to have a higher risk of developing sleep disorders, including insomnia [51]. Furthermore, a study conducted among 159,000 US respondents found that poor socioeconomic situations and a lower level of education increase the risk of developing sleep disorders [52]. In the case of Korean adolescents, it was found that the lower the average household income, the worse the sleep quality was [53]. These data are consistent with the results of this study, in which students in the richest countries, according to the GDP per capita index, showed a significantly lower risk of developing insomnia (OR 0.45; *p* < 0.001) and excessive daytime sleepiness (OR 0.74; *p* = 0.008). Determining the causes of this phenomenon is not simple, and the available data are not conclusive. According to the authors of this study, this may be due to lifestyle. Those with a higher level of education and a higher income often pay more attention to health-promoting behaviors, including sleep hygiene [31,32,52]. Another reason may be student housing conditions. Poor housing and neighborhood conditions were found to contribute to poorer sleep quality and duration [54]. It was also found that those from lower-income countries are more likely to suffer from mental disorders such as anxiety and depression, which have a huge impact on sleep quality. Furthermore, it is important to bear in mind that health services, diagnostic evaluation, and the treatment of chronic conditions are of lower quality in lower-income countries, and this undoubtedly affects sleep quality.

When analyzing the results, it is also important to bear in mind the ongoing COVID-19 pandemic and its enormous impact on sleep. For example, in a study regarding the subjective assessment of the impact of the COVID-19 pandemic on sleep, as many as 94.9% of students clearly indicated that the pandemic had affected their sleep [55]. So far, it has been proven that the pandemic led to changes in sleep habits among students: going to bed later, getting out of bed later, having more naps during the day, and increased sleep duration [16,56]. Such behavior results in changes to the diurnal rhythm and worsening sleep quality despite a relative increase in its duration. This phenomenon may explain the results of this study, in which the pandemic period increased the risk of developing insomnia by 2.17 times with a decrease in the risk of excessive daytime sleepiness (OR 0.78; *p* < 0.001). It should also be kept in mind that human mental health has deteriorated due to the ongoing pandemic, which is in close correlation with sleep quality [57].

This study revealed that nonmedical students had a higher risk of developing insomnia, while medical students were more likely to suffer from excessive daytime sleepiness. These findings appear to contradict previous reports, in which it was future medics who were more likely to report sleep problems in all forms. A Lithuanian study found that medical students were significantly more likely to have sleep problems compared to law students or economics students [58]. Similar observations were made by comparing law students with medical students in India [59]. According to a study by Sharma et al., medical students scored on average 1.27 points higher on the PSQI (Pittsburgh sleep quality index) scale than other students, and 60% reported insufficient sleep duration [60]. On the other hand, the significantly higher risk of excessive daytime sleepiness among medical students may be due to the heavy burden of study, vocational training practice, including night duty, and high mental workload, and this was proved in observations among students in, among others, Malaysia, Brazil or Jordan [16,17,55].

In contrast, the results of this study appear to be consistent with previous reports, in which it was found that first-year university students have a higher risk of developing sleep disorders, and this may be due to both sudden life changes, such as the start of university and relocation, or new social relationships. Increased evening activity and the more frequent consumption of stimulants may be other potential causes of sleep disorders [61,62,63]. However, it is important to bear in mind that the effects of stimulants on sleep are very complex, and studies to date do not provide a definitive answer. Although this study did not prove that the use of alcohol within the last three months increased the risk of insomnia or daytime sleepiness, its long-term use may contribute to prolonged sleep latency and frequent awakenings, which significantly reduces sleep quality [64]. In the case of the use of psychostimulants, the results of this study clearly show their negative effects on sleep, contributing to an increased risk of both insomnia and daytime sleepiness. A study conducted among US students found that psychostimulant users reported significantly lower sleep quality [65]. In contrast, a study in animal models confirmed that the use of psychostimulants alters sleep architecture [66]. It must also be kept in mind that the relationship between sleep and stimulants is a two-way relationship. On the one hand, the use of stimulants may lead to changes in sleep patterns and deterioration of sleep quality. On the other hand, the presence of sleep disorders significantly increases the risk of using stimulants [7].

Another parameter that was measured in this study was quality of life (QoL), which is not the easiest to measure. Due to its multifaceted nature, the objective assessment of QoL (AQoL) requires the use of appropriately designed psychometric tools. According to the authors, the questionnaire used in this study addresses many aspects of life that affect the sense of QoL, with high sensitivity and specificity [38,39]. This study revealed that students scored, on average, 60.9 ± 11.46 points in the MANSA questionnaire. The presence of sleep disorders, both in the form of insomnia and daytime sleepiness, was found to significantly impair students’ QoL. These reports are consistent with worldwide observations in which abnormal sleep significantly reduced the QoL of students in Iran, Portugal, and Rome [67,68,69]. The effect of sleep on QoL is also relevant in population-based assessments. S. Lee’s study, which was conducted among 225,541 Koreans, revealed that poor sleep quality significantly reduces QoL, particularly when anxiety or depressive disorders are also present [70]. In contrast, the analysis of the individual questions of the scale revealed that inadequate sleep significantly reduced the satisfaction of students’ mental and physical health, which is also consistent with Korean observations, according to which those with poor sleep quality had problems with regular physical activity [70]. In addition to sleep quality, there was also a correlation between the HDI of students’ place of residence and the MANSA total score, according to which students in countries with a very high HDI obtained the highest MANSA scores. This is also consistent with previous data, i.e., the higher a country’s development status, the higher the level of social prosperity, which improves the QoL of its inhabitants [71].

The authors are aware of the limitations of this study, which undoubtedly include the methodology for data collection using an anonymous online questionnaire. The use of such a method, however, makes it possible to reach a very large audience from all over the world and increases the chance of participation in the survey. Previous studies found that anonymous online surveys strongly reduce feelings of anxiety among respondents and increase the likelihood of participation [72]. On the other hand, the authors of anonymous online surveys have no possibility of verifying the veracity of the data left and determining the response rate. Moreover, the used psychometric tools are based on the subjective assessment of respondents and need to be objectified through psychiatric examination. It should also be mentioned that the software used prevented the detection of bots. The interpretation of this study’s findings is also limited by the fact that the authors did not collect data regarding chronic conditions, the chronic medications used, and the mental health conditions that may potentially be present in the surveyed population. Another limitation of this study is the great disparity between the respondents in terms of HDI, GDP per capita, and sex, with a marked predominance of women. It should be noted, however, that the used psychometric tools are distinguished research methods. Furthermore, the significant size of the study group from 60 countries around the world and the observation period covering both the COVID-19 pandemic period and the period before its outbreak contribute to the strength and innovation of this study.

In conclusion, the results of this study clearly indicate that sleep disorders are a serious health problem among young adults, which has been further exacerbated by the ongoing COVID-19 pandemic. However, further observations on representative groups of students in countries with different levels of development and wealth are necessary to better understand the phenomenon of insomnia in relation to socioeconomic status. In addition, it could be useful to prepare relevant meta-analyses using existing data.

Solutions to this problem should also be sought, which could be represented by the implementation of appropriate social campaigns on proper sleep hygiene and the importance of sleep for health, as well as appropriate education on the use of stimulants and their impact on health and sleep. For this purpose, it would be possible to use the mass media, with enormous potential. In addition, it would be possible to implement appropriate classes on healthy lifestyles as one of the possible subjects during education.

## 5. Conclusions

Insomnia and excessive daytime sleepiness are common conditions among students worldwide and are closely related. The COVID-19 pandemic significantly altered students’ diurnal rhythms, which has contributed to an increase in insomnia. Students in countries with a high GDP per capita index are significantly less likely to develop insomnia compared to the residents of countries with a low GDP per capita index. Sleep disorders definitely reduce the QoL of students.

## Figures and Tables

**Table 1 jcm-12-00692-t001:** Characteristics of the study group for the total population and by the COVID-19 pandemic.

Variable		Entire Study Group N (%)	Before the Pandemic N (%)	During the Pandemic N (%)	Size Effect	*p* ^^^
Sex	Female	15,743 (78.2)	7887 (77.4)	7856 (78.9)	6.69 *	**0.010**
	Male	4396 (21.8)	2300 (22.6)	2096 (21.1)		
Age M ± SD		22.6 ± 3.6	22.4 ± 3.0	22.7 ± 4.1	0.083 ^#^	**<0.001**
Year of study	I	5504 (27.3)	2180 (21.4)	3324 (33.4)	458.52 *	**<0.001**
	II	3600 (17.9)	2106 (20.7)	1494 (15.0)		
	III	3761 (18.7)	2173 (21.4)	1588 (16.0)		
	IV	3119 (15.5)	1624 (15.9)	1495 (15.0)		
	V	2885 (14.3)	1530 (15.0)	1355 (13.6)		
	VI	1270 (6.3)	574 (5.6)	696 (7.0)		
University profile	Medical	4518 (22.4)	2578 (25.3)	1940 (19.5)	97.73 *	**<0.001**
	Non-medical	15,621 (77.6)	7609 (74.7)	8012 (80.5)		
HDI	Very high	18,206 (90.4)	9432 (92.6)	8774 (88.2)	146.77 *	**<0.001**
	High	785 (3.9)	364 (3.6)	421 (4.2)		
	Medium	1000 (5.0)	329 (3.2)	671 (6.7)		
	Low	148 (0.7)	62 (0.6)	86 (0.9)		
GDP per capita	High	17,716 (87.9)	9252 (90.8)	8464 (85.0)	175.27 *	**<0.001**
	Upper-middle	1059 (5.3)	455 (4.5)	604 (6.1)		
	Lower-middle	973 (4.9)	348 (3.4)	625 (6.3)		
	Low	391 (1.9)	259 (2.6)	132 (1.3)		
COVID-19 pandemic announcement	Before the pandemic	10,187 (50.6)	---	----	---	---
	During the pandemic	9952 (49.4)	---	---		
Place of study	Europe	17,954 (89.1)	9372 (92.1)	8582 (86.2)	220.82 *	**<0.001**
	North America	147 (0.7)	65 (0.6)	82 (0.8)		
	South America	173 (0.9)	65 (0.6)	108 (1.1)		
	Asia	1425 (7.1)	474 (4.7)	951 (9.6)		
	Africa	390 (1.9)	167 (1.6)	223 (2.2)		
	Australia	50 (0.3)	35 (0.3)	15 (0.2)		
Alcohol	Yes	16,517 (82.0)	8833 (86.8)	7684 (77.1)	317.47 *	**<0.001**
	No	3622 (18.0)	1345 (13.2)	2277 (22.9)		
Cannabinoids	Yes	2731 (13.6)	1449 (14.2)	1282 (12.9)	8.01 *	**0.004**
	No	17,408 (86.4)	8729 (12.9)	8679 (87.1)		
Psychostimulants	Yes	729 (3.6)	432 (4.2)	297 (3.0)	23.01 *	**<0.001**
	No	19,410 (96.4)	9746 (95.8)	9664 (97.0)		
Sedatives	Yes	2522 (22.6)	1183 (11.6)	1339 (13.4)	15.21 *	**<0.001**
	No	17,617 (87.4)	8995 (88.4)	8622 (86.6)		

M—mean; SD—standard deviation; HDI—Human Development Index; GDP—Gross domestic product; * Chi^2^; ^#^ Cohen’s d; ^^^ comparison of distribution in relation to the COVID-19 pandemic. Statistically significant values are in bold with the significance level set at *p* < 0.05.

**Table 2 jcm-12-00692-t002:** The backward stepwise logistic regression model assessment of the effects of sociodemographic variables, university profile, pandemic, place of residence, and psychoactive substance use on insomnia and daytime sleepiness.

Variable		AIS				ESS			
		OR	95%Cl	*p*	NG	OR	95%Cl	*p*	NG
Age M ± SD		1.06	[1.05, 1.07]	**<0.001**	0.113	1.01	[1.00, 1.02]	**<0.001**	0.017
During the pandemic		2.17	[2.05, 2.31]	**<0.001**		0.78	[0.74, 0.84]	**<0.001**	
Sex	F	1.25	[1.67, 1.35]	**<0.001**		1.44	[1.33, 1.56]	**<0.001**	
Year of study	VI	0.87	[0.75, 1.01]	**0.073**		0.82	[0.70, 0.95]	**0.007**	
	V	0.82	[0.73, 0.91]	**<0.001**		0.77	[0.68, 0.87]	**<0.001**	
	IV	0.88	[0.73, 0.97]	**0.012**		0.82	[0.74, 0.91]	**<0.001**	
	III	0.97	[0.89, 1.07]	0.661		0.89	[0.81, 0.98]	**0.017**	
	II	1.14	[1.04, 1.25]	**0.003**		0.95	[0.87, 1.05]	0.362	
University profile	Medical	0.90	[0.84, 0.97]	**0.004**		1.20	[1.12, 1.29]	**<0.001**	
HDI	Very high	1.41	[0.74, 2.67]	0.297		---	---	---	
	High	1.14	[0.64, 2.06]	0.647		---	---	---	
	Medium	1.70	[1.04, 2.76]	**0.033**		---	---	---	
GDP per capita	High	0.41	[0.24, 0.69]	**<0.001**		0.74	[0.59, 0.93]	**0.008**	
	Upper-middle	0.76	[0.47, 1.25]	0.287		1.02	[0.79, 1.32]	0.875	
	Lower-middle	0.98	[0.68, 1.42]	0.928		1.06	[0.82, 1.37]	0.644	
Alcohol	Yes	---	---	---		---	---	---	
Cannabinoids	Yes	1.10	[1.00, 1.20]	**0.034**		---	---	---	
Psychostimulants	Yes	1.51	[0.63, 1.89]	**<0.001**		1.14	[1.04, 1.26]	**0.003**	
Sedatives	Yes	2.60	[2.36, 2.87]	**<0.001**		1.27	[1.16, 1.39]	**<0.001**	

HDI—human development index; GDP—gross domestic product; AIS—Athens insomnia scale; ESS—Epworth sleepiness scale; NG—Nagelkerke pseudo-R^2^; OR—odds ratio; 95%CI—confidence interval of OR; *p*—statistical significance; statistically significant values are in bold with the significance level set at *p* < 0.05.

**Table 3 jcm-12-00692-t003:** The effects of sociodemographic variables, pandemic, university profile, year of study, insomnia, and daytime sleepiness on the subjective AQoL in a complex backward stepwise linear regression model.

Variable			MANSA				
		B	β	SE	t	*p*	NG
During the pandemic		−0.406	−0.035	0.078	−5.18	**<0.001**	0.110
University profile	Medical	1.124	0.081	0.091	12.30	**<0.001**	
HDI	Very high	2.001	0.058	0.656	3.04	**0.002**	
	High	0.105	0.002	0.490	0.21	0.829	
	Medium	−1.795	−0.040	0.486	−3.70	**<0.001**	
GDP per capita	High	−0.874	−0.031	0.565	−1.54	0.122	
	Upper-middle	0.767	0.017	0.443	1.73	0.083	
	Lower-middle	−0.091	−0.002	0.453	−0.20	0.840	
Year of study	VI	0.711	0.033	0.263	2.69	**0.007**	
	V	0.386	0.021	0.182	2.12	**0.033**	
	IV	0.126	0.007	0.176	0.72	0.474	
	III	−0.273	−0.016	0.165	−1.65	0.097	
	II	−0.121	−0.007	0.167	−0.72	0.469	
Alcohol	Yes	0.563	0.038	0.107	5.24	**<0.001**	
Cannabinoids	Yes	−0.260	−0.015	0.116	−2.25	**0.024**	
Psychostimulants	Yes	−0.629	−0.021	0.210	−2.298	**0.002**	
Sedatives	Yes	−2.357	−0.137	0.114	−20.54	**<0.001**	
AIS	Insomnia	−3.142	−0.271	0.079	−39.41	**<0.001**	
ESS	Daytime sleepiness	−1.331	−0.103	0.086	−15.56	**<0.001**	

HDI—human development index; AIS—Athens insomnia scale; ESS—Epworth sleepiness scale; NG—Nagelkerke pseudo-R^2^; B—coefficient value of a given variable; SE—standard error; t—test value, *p*—statistical significance; statistically significant values are in bold with the significance level set at *p* < 0.05.

## Data Availability

The data presented in this study are available on request from the corresponding author.

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
