# Peer review of "Insomnia, Daytime Sleepiness, and Quality of Life among 20,139 College Students in 60 Countries around the World—A 2016–2021 Study"

_jcm, 2023, doi:10.3390/jcm12020692_

Round 1
Reviewer 1 Report
The manuscript reports an interesting study about sleep quality and quality of life in a large sample of university students. The data is well presented and the methods are clear. I do not have specific concerns, however, I think the authors should consider some aspects for the improvement of their manuscript:
- Be sure to introduce all acronyms before their use—for example, GDP in the abstract and line 110.
- I think that for reproducibility the authors should report the rates of the responders based on the countries (or at least big areas like West Europe, East Europe, North America, etc).
- It is not clear to me if the p-values reported in Table 1 are related to the comparison between subgroups (for example male vs female) or between COVID-19 periods. Please clarify. Moreover, if the p-values referred to COVID-19 periods, have you evaluated to use of this data as a covariate for the main analyses?
- Have you used any method to avoid bot responders?
Author Response
Thank you sincerely for your time in reviewing our article. In response to your suggestions, we have added the following information as part of our work
1) The acronyms used were explained before their first use in the text.
2) Table 1 has been enriched with data on the place of study (continent). In addition, the table description clarified what the p-value refers to. As for the analysis, the stage of the study (before/ during the pandemic) was included in each multivariate model.
3) Unfortunately, the portal we use does not provide the possibility of blocking bot activity, among other things, 1 survey for 1 IP address. We added the information as part of the survey limitation.
I hope that in its current form in your opinion this article meets the publication criteria.
Reviewer 2 Report
dear authors, thank you for the submission.
this is a very well written manuscript and well presented two minor comments to add:
1) provide better interpretation for effect sizes
2) provide paragraph about future rcts in this populations and need for meta-analyses
Author Response
Dear Reviewer,
Thank you sincerely for your time in reviewing our article. In response to your suggestions, we have added information in the discussion summary about the need for further studies on representative groups and the preparation of meta-analyses using available data.
I hope that in its present form in your opinion this article meets the publication criteria.